evolution/genomics

$Q_{ST}$-$F_{ST}$, natural selection, microsatellite, quantitative genetics, *Pungitius pungitius*, single-locus polymorphisms

**Author for correspondence:**
Antoine Fraimout
e-mail: antoine.fraimout@helsinki.fi

†These authors contributed equally to this work.

# Effects of marker type and filtering criteria on $Q_{ST}$-$F_{ST}$ comparisons

Zitong Li[1,†], Ari Löytynoja[2,†], Antoine Fraimout[1] and Juha Merilä[1]

[1]Ecological Genetics Research Unit, Organismal and Evolutionary Biology Research Programme, and [2]Institute of Biotechnology, University of Helsinki, Helsinki 00014, Finland

AF, 0000-0003-4552-3553; JM, 0000-0001-9614-0072

Comparative studies of quantitative and neutral genetic differentiation ($Q_{ST}$-$F_{ST}$ tests) provide means to detect adaptive population differentiation. However, $Q_{ST}$-$F_{ST}$ tests can be overly liberal if the markers used deflate $F_{ST}$ below its expectation, or overly conservative if methodological biases lead to inflated $F_{ST}$ estimates. We investigated how marker type and filtering criteria for marker selection influence $Q_{ST}$-$F_{ST}$ comparisons through their effects on $F_{ST}$ using simulations and empirical data on over 18 000 *in silico* genotyped microsatellites and 3.8 million single-locus polymorphism (SNP) loci from four populations of nine-spined sticklebacks (*Pungitius pungitius*). Empirical and simulated data revealed that $F_{ST}$ decreased with increasing marker variability, and was generally higher with SNPs than with microsatellites. The estimated baseline $F_{ST}$ levels were also sensitive to filtering criteria for SNPs: both minor alleles and linkage disequilibrium (LD) pruning influenced $F_{ST}$ estimation, as did marker ascertainment. However, in the case of stickleback data used here where $Q_{ST}$ is high, the choice of marker type, their genomic location, ascertainment and filtering made little difference to outcomes of $Q_{ST}$-$F_{ST}$ tests. Nevertheless, we recommend that $Q_{ST}$-$F_{ST}$ tests using microsatellites should discard the most variable loci, and those using SNPs should pay attention to marker ascertainment and properly account for LD before filtering SNPs. This may be especially important when level of quantitative trait differentiation is low and levels of neutral differentiation high.

## 1. Introduction

Geographical and temporal differentiation in mean values of quantitative traits are of commonplace occurrence in animal and plant populations. Before such differentiation can be ascribed to adaptation, two premises need to be fulfilled. First, the observed

differentiation needs to be genetically based rather than environmentally induced. Second, the observed differentiation has to have been caused by directional natural selection instead of neutral processes, such as gene flow or genetic drift. Comparative studies of quantitative trait and molecular marker differentiation ($Q_{ST}$-$F_{ST}$ tests) provide a way of identifying footprints of directional selection as a cause of population differentiation in quantitative traits (reviewed in [1–5]). The rationale behind $Q_{ST}$-$F_{ST}$ comparisons is simple: the level of genetically based population differentiation in quantitative traits (i.e. $Q_{ST}$) is compared with that in neutral marker loci (i.e. $F_{ST}$). If $Q_{ST} > F_{ST}$, this provides evidence for adaptive population differentiation, as only directional selection is expected to elevate divergence in quantitative traits above the neutral expectation. Similarly, if $Q_{ST} < F_{ST}$, this would be indicative of uniform stabilizing selection that has prevented populations from diverging less than would be expected under random genetic drift alone. An outcome of $Q_{ST} \approx F_{ST}$ would be inconclusive: the null-hypothesis of differentiation by genetic drift could not be rejected (e.g. [1]). While the majority of studies using the $Q_{ST}$-$F_{ST}$ (reviewed in [3,4,6]) or a related approach [7,8] to test for the adaptive nature of quantitative trait differentiation have found support for it, there is a concern that many of these studies might have tested $Q_{ST}$ against overly liberal (i.e. too low) $F_{ST}$ estimates [9,10]. This concern stems from the fact that most $Q_{ST}$-$F_{ST}$ comparisons have used microsatellite markers for $F_{ST}$ estimation. Since microsatellite loci are highly variable due to their high mutation rates [11,12], they may underestimate levels of neutral genetic differentiation [13–16], and hence, render $Q_{ST}$-$F_{ST}$ comparisons biased towards finding evidence for adaptive differentiation [9,10]. Recent advances in sequencing technology have started to replace microsatellite markers with single-locus polymorphisms (SNPs) as the tool of the trade-in population genetic investigations of non-model organisms [17–19]. Since SNP loci are (mostly) bi-allelic and experience, on average, lower mutation rates than microsatellite loci, they might provide a way to obtain less biased estimates of neutral baseline differentiation than microsatellite loci [9,20]. However, since per nucleotide mutation rates are highly variable, and may vary depending on their genomic location [21,22], SNP loci may not automatically yield less biased neutral baseline estimates of divergence (as measured by $F_{ST}$) than microsatellites. Similarly, not all SNPs are or behave as being neutral; they may be subject to various forms of selection depending on their genomic location and/or functionality [23–25]. For instance, hitchhiking under positive and background selection can increase allele frequency shifts between populations, especially in genomic regions of low recombination [26–28]. Bias in the estimation of the neutral baseline from SNP data could also arise from analytical methods involved in marker selection and $F_{ST}$ estimation [29]. Specifically, different averaging methods to estimate $F_{ST}$ across loci have been shown to yield different results between studies [29], which should be particularly concerning in the context of $Q_{ST}$-$F_{ST}$ comparisons. On the other hand, $F_{ST}$ estimates can be affected by ascertainment bias (*sensu* [17]) when the selected markers are not representative of the variability observed in all sampled individuals. They can also be sensitive to how rare low-frequency alleles are collected and filtered from the data [30–33], and how each researcher decides to deal with markers that are in linkage disequilibrium (LD) with each other [34]. The latter issue may be particularly important in $Q_{ST}$-$F_{ST}$ studies where divergent populations differing in their LD patterns are compared: due to methodological constraints, typically, only a small subset of SNPs can be included and failure to account for LD in marker selection for one or more populations in the data has potential to bias $F_{ST}$ estimates. To the best of our knowledge, there have not been any studies of these effects of SNPs filtering in the context of $Q_{ST}$-$F_{ST}$ comparisons, although the problem of defining neutral baseline differentiation level has been well recognized in related contexts (e.g. [25,35,36]).

The aim of this study was to explore how marker type and various filtering criteria could influence inferences from $Q_{ST}$-$F_{ST}$ comparisons through their effect on estimates of neutral baseline differentiation. To this end, we used empirical data on quantitative trait and microsatellite differentiation among four populations of nine-spined sticklebacks (*Pungitius pungitius*) used in an earlier $Q_{ST}$-$F_{ST}$ study [8]. These data were supplemented with information on variability in millions of SNP loci and over 18 000 *in silico* genotyped microsatellite loci to see how marker type and different filtering criteria influenced the neutral baseline differentiation level as reflected in $F_{ST}$. In particular, we were interested to evaluate the conjecture (cf. [10]) that $Q_{ST}$-$F_{ST}$ comparisons using highly variable microsatellite markers could be biased towards finding evidence for adaptive differentiation. Because the *P. pungitius* study system is known for its particularly high level of quantitative trait differentiation [8,37,38], we also simulated genomic datasets under varying levels of $Q_{ST}$-$F_{ST}$ divergence and evaluated the different marker types' ability to detect the signature of divergent selection under different evolutionary scenarios in this particular system.

# 2. Material and methods

## 2.1. Phenotype data

The phenotype data used here were adopted from an earlier study [8] which used a full-sib mating design comprised of 92 $F_1$-generation nine-spined stickleback individuals from four populations (Baltic Sea, $n = 21$; White Sea, $n = 24$, Bynastjärnen, $n = 22$, Pyöreälampi, $n = 25$) and five families per population. Five morphological traits including standard length (M1), body depth (M2), head length (M3), pelvic girdle length (M4) and caudal peduncle length (M5), as well as two behavioural traits including aggressiveness (B1) and propensity for risk-taking (B2) of all 92 individuals were quantified as described in Herczeg *et al.* [37,38]. The study populations are highly differentiated in most of these traits, and the available evidence suggested that this differentiation has been driven by natural selection stemming from local adaptation to pond environments (Bynastjärnen & Pyöreälampi) lacking piscine predators [8,37–39].

## 2.2. Genetic data

An assembled and annotated reference genome (110× coverage) for *P. pungitius* from a Pyöreälampi individual [40] was used for obtaining the genetic data. SNP data and *in silico* microsatellite data were obtained by whole genome resequencing of 10 female individuals from each of the above-mentioned four populations. The processing and analysis of these are explained in detail below. The empirical microsatellite data for 12 loci were the same as that used by Karhunen *et al.* [8].

## 2.3. Variant calling

The 40 samples for this study were processed as a part of a larger sample set ($n = 140$) that included populations not used in this study. For each sample, the sequence reads were mapped to the reference genome with bwa (v. 0.7.12; [41]) using default options, duplicate reads were removed with SAMtools (v. 1.3.1; [42]) and those around indels were realigned with GenomeAnalysisToolkit's IndelRealigner (GATK; v. 3.4; [43]). Each sample was then called with GATK HaplotypeCaller using mode 'discovery', output format 'gvcf', and emit and call confidence of 3 and 10, respectively. The full sample set was jointly genotyped with GATK GenotypeGVCFs. From this, the 40 samples from the four study populations were extracted and their analysis continued in isolation.

## 2.4. SNP filtering

The full datasets consist of 3 806 181 SNPs that are variable within the four populations and have data for at least 80% of the individuals in each population. We applied different SNP filtering criteria to obtain the following three datasets: (i) SNPs in all genomic regions (ALL), (ii) the SNPs in non-coding regions excluding exons and repeat regions and their immediate neighbourhood (50 bp of flanking sequence) (NONCOD) and (iii) the SNPs in synonymous third codon position (C3SYNO). C3SYNO includes variable third codon positions that were inferred to contain synonymous substitutions by the R package VariantAnnotation (v. 1.18.1; [44]). From these, we removed sites with missing genotype data or monomorphic among the study samples and the three final datasets consisted of 1 702 105, 1 165 221 and 41 927 SNPs, respectively. To account for the possibility of ascertainment bias [17,29], we also filtered markers based on their variability in the marine populations. This is because $F_{ST}$ can be seen as either a parameter of the evolutionary process, or a statistic derived from observed samples. As such, it is used to measure correlation between randomly drawn alleles from a single population relative to either the most recent common ancestral population or the combination of the two population samples [29]. Under the former definition, the markers included in the analysis should have been variable in the ancestral population; alternatively, markers can often be ascertained by their variability in an outgroup or maximally diverged modern populations. In our case, the marine populations are representative of the ancestral genetic variation and represent the geographical extremities among the study populations. Hence, variability-based marker ascertainment was done using the two marine populations and for each category, markers invariable in either of the marine populations were excluded.

## 2.5. Effect of LD on $F_{ST}$

Filtering of SNP dataset based on the degree of linkage between loci is a common pruning method in population genomics studies. For $Q_{ST}$-$F_{ST}$ comparison, no clear guidelines are available on whether SNPs in LD should or should not be included in the data used for $F_{ST}$ estimation. A general assumption is that unlinked markers should be preferred as they provide a closer estimation of what would be expected under neutrality. For practical and computational reasons, LD pruning should at least be performed on the basis that some $Q_{ST}$-$F_{ST}$ methods—particularly MCMC-based approach—cannot handle large numbers of markers. Consequently, we investigated the effect of LD pruning on the estimation of $F_{ST}$ by using genome-wide sets of unlinked SNPs to estimate the mean and variance of $F_{ST}$. Two different approaches were used to extract a subset of unlinked SNPs from the full SNP datasets. First, a thinning approach was applied to select roughly 2000 SNPs located at *ca* 150–200 kb from each other, and the rest of the SNPs were discarded. The thinning approach was carried out with VCFtools [45]. Second, we used the more sophisticated LD pruning approach of Zheng *et al.* [46] implemented in the R package *SNPrelate*. This approach divides each chromosome first into multiple 500 kb non-overlapping sliding windows. Within each sliding window, a single SNP was firstly randomly included into the active set, and this was followed by including another SNP into the active set if its correlation (as defined by the LD composite measure; [47,48]) with the SNPs in the active set was lower than a defined threshold. This procedure was repeated for all the SNPs in the same sliding window. A sequence of LD-values ranging from 0.05 to 0.95 (the pruned data have low LD when the threshold is small) were used to evaluate how the $F_{ST}$ estimates were affected by the changes of LD. To see how the thinning and sliding window approaches influenced the baseline $F_{ST}$ levels as estimated from the data, we evaluated $F_{ST}$ as a function of standardized (to the total number of markers in the data) marker number included into the estimation. If the pruning is effective, we would expect no relationship between $F_{ST}$ and number of markers included into the estimation. However, if the pruning is ineffective and linked markers become included into the data, we expect to see that the $F_{ST}$ varies as a function of number of markers included. Comparison of LD between pond and marine populations was done by calculating the mean LD (defined as squared composite measure; [47,48]) between all the SNP pairs within each non-overlapping 500 kb sliding window across the genome.

## 2.6. *In silico* microsatellite analysis

The *in silico* microsatellite discovery and genotyping was performed with Tandem repeats finder (TRF; v. 4.09; [49]) and RepeatSeq (v. 0.8.2; [50]), respectively. Dinucleotide repeats were identified in the *P. pungitius* reference genome using TRF with options '2 7 7 80 10 20 2 -f -d -m -h'. The parameter values are as recommended by the software developers with the exception of the last two, minimum score (20) and maximum period size (2). From the output, repeat loci were chosen with the following criteria: period and consensus pattern of 2, length of 10–20 copies, alignment score of 100 and distance of at least 500 bp to the previous locus. The selected loci were genotyped in the 40 study samples using RepeatSeq. The output was processed using bcftools [42] and custom scripts, and the resulting data analysed using the R package diveRsity (v. 1.9.89; [51]). After removing loci with more than 20% missing data in any population, the resulting dataset contained 18 824 microsatellite loci.

## 2.7. Estimation of local recombination rate

Since variation in recombination rate along chromosomes can influence patterns of polymorphism and hence $F_{ST}$ through background and/or hitchhiking selection (e.g. [28,52]), we explored how the $F_{ST}$ in our data was influenced by variation in recombination rate. To this end, the local recombination rate was estimated using a sex-averaged genetic map (an improved [40] version of that reported in [53]). First, non-monotonic regions were manually removed and outlier markers discarded by performing loess regression ($\alpha = 0.2$, degree = 1), interpolating genetic distance from physical positions and removing any points that lie further than 2 cM from the line. Using the remaining regions and markers, we estimated the local recombination rate by performing loess regression on a fixed number of markers ($\alpha$ adjusted to include 250 markers, degree = 2) and predicting the change on genetic distance for a pair of adjacent physical positions. We could estimate the local recombination rate for 54.7% of the genome (electronic supplementary material, figure S1). Although a small fraction of the genome still got negative rate estimates, a great majority of sites had inferred recombination rates between 0 and 20 cM Mb$^{-1}$ (mean = 6.6). Using the inferred rate, the genome

was split into two roughly equally large subsets of regions with low (1–7 cM Mb$^{-1}$) and high (7–15 cM Mb$^{-1}$) local recombination rates, discarding the loci with either no rate estimate or an estimate from the extreme ends of the rate distribution (electronic supplementary material, figure S2a). The microsatellite loci were further divided into those with few (1–4), intermediate (5–8) and many (9–21) alleles, the two extremes approximately corresponding to the first and fourth quartile of the distribution (electronic supplementary material, figure S2b). $F_{ST}$ was separately estimated for the different subsets of SNP dataset NONCOD and the microsatellite data. Since the impact of background selection was expected to be weak in small pond populations subjected to strong drift, the impact of recombination rate on $F_{ST}$ was assessed only in the two large marine populations (LEV & HEL).

## 2.8. Outlier analyses

Since loci linked to selected sites are expected to show elevated $F_{ST}$ [36], we used two approaches to identify possible outliers and evaluate their impact on baseline $F_{ST}$: BayeScan [54] and OutFLANK [33]. In both methods, the outliers were detected while controlling for the false discovery rate with a liberal $p = 0.1$ threshold.

## 2.9. $F_{ST}$ estimation and comparison

We investigated whether different estimation methods would yield critically different $F_{ST}$ values as expected from previous studies [29]. First, we used the Weir & Cockerham [55] theta estimator of $F_{ST}$. This estimator can be efficiently obtained even from genome-wide datasets, and it was used for estimation of both single and multilocus $F_{ST}$. This estimator was used to (i) calculate the pairwise $F_{ST}$ values for each individual marker and produce empirical $F_{ST}$ distributions for different SNP (using VCFtools; [45]) and microsatellite datasets (R package *diveRsity*; [51]), and (ii) evaluate how different ways of filtering and pruning the SNP data can affect the estimation of the neutral baseline $F_{ST}$ (using R package *SNPrelate*; [46]). We then combined the estimates across loci using either the 'ratio of average' (ROA) or the 'average of ratio' (AOR) [29].

## 2.10. $Q_{ST}$-$F_{ST}$ comparisons

We used two approaches to conduct the $Q_{ST}$-$F_{ST}$ comparisons: the Bayesian Driftsel [7] and the frequentist QstFstComp approach [56,57]. One advantage of the former over the latter approach is that it has been shown to be more powerful in detecting signatures of selection even when the number of populations is low (see [58] for a detailed statistical demonstration). However, since the standard $Q_{ST}$-$F_{ST}$ comparisons continue to be widely used, and because they might be more intuitive in providing direct comparison of the quantities of $F_{ST}$ and $Q_{ST}$, QstFstComp tests were included for comparison. Driftsel provides two test statistics, referred to as S and H statistics. S statistic accounts for patterns of relatedness among populations, as well as ancestral genetic correlations among the traits of interest [58]. S-values close to zero are indicative of stabilizing selection; those close to one indicate directional selection; and values close to 0.5 are consistent with evolution due to drift [58]. In addition to the factors accounted for with the S statistic, the H statistic allows the environment to be accounted for by including similarity in the distribution of population means and habitat parameters [8]. In other words, the H statistics allows one to test whether population means from similar habitats are more similar than expected by random genetic drift. Following the testing criteria proposed in Karhunen *et al.* [8], S or H > 0.95 implies that a quantitative trait has evolved under divergent selection at the 95% credibility level, whereas S or H < 0.05 would imply stabilizing selection at the same credibility level. The default non-informative priors were used in the Driftsel analyses. 15 000 Markov chain Monte Carlo (MCMC) samples of the posterior distribution were simulated, by considering the first 5000 as a burn-in, and the remaining were stored in every 10th iteration, so that eventually 1000 MCMC samples were used for calculating S and H statistics. We calculated both statistics for all datasets, using the binary habitat type (pond versus marine) as the input data for the distance matrix of the environmental covariates required by Driftsel [8]. The frequentist QstFstComp constructs a null distribution of the difference between $Q_{ST}$ and $F_{ST}$ using a parametric simulation approach. The sampling error of the $F_{ST}$ was evaluated by random sampling (i.e. bootstrapping) of the given loci, and the expected distribution of the $Q_{ST}$ under neutrality was simulated on the basis of the mean $F_{ST}$. The observed $Q_{ST}$-$F_{ST}$ quantity was compared with 1000 samples simulated from the $Q_{ST}$-$F_{ST}$ null distribution for a statistical test of neutrality, from where

uncertainty quantities ($p$-values and confidence intervals) were derived. If $Q_{ST}$-$F_{ST}$ quantity is positive, and $p < 0.05$, the quantitative trait in question is inferred to have been subject to divergent selection. The QstFstComp was not applicable to the microsatellite loci with more than 10 alleles due to software limitation. Since both methods, and in particular the MCMC-based Driftsel approach, are computationally very demanding, we restricted all the analyses to 2000 loci. For SNP analyses, SNP loci obtained with the thinning approach (see above) were derived from the ALL, NONCOD and C3SYNO datasets. Likewise, loci for *in silico* microsatellite analyses were obtained with the thinning approach, and the analyses were conducted separately for all loci irrespectively of their variability, only loci with a low number of alleles, and only loci with a high number of alleles. Finally, we investigated the effect of marker ascertainment (cf. *SNP filtering* section above and [29]) on $Q_{ST}$-$F_{ST}$ comparisons. For each genomic dataset (i.e. ALL, NONCOD and C3SYNO) and each set of *in silico* microsatellites, we ran Driftsel using full datasets and using loci ascertained based on their variability in the marine populations. QstFstComp was also run for each genomic dataset using unascertained and ascertained SNP markers.

## 2.11. Simulation study

A simulation study was conducted to evaluate the ability of Driftsel and QstFstComp to detect signatures of divergent selection with different marker types and varying levels of $Q_{ST}$-$F_{ST}$ divergence. Because our empirical data show a high level of quantitative trait differentiation, the rationale here was to simulate different levels of $Q_{ST}$-$F_{ST}$ divergence to evaluate the effect of marker types on $Q_{ST}$-$F_{ST}$ inference in a more general context. To do so, a population genomic dataset of four sub-populations corresponding to our four study populations (Helsinki [HEL], White Sea [LEV], Bynastjärnen [BYN] and Pyöreälampi [PYO]) was first simulated using the software *fastsimcoal2* [59,60]. Simulations were based on the most likely demographic history of real nine-spined stickleback populations (electronic supplementary material, figure S3 and see [61–63]): first, a population split was modelled between the two marine (LEV and HEL) sub-populations from an ancestral population 5000 and 4000 generations ago, respectively. Then the pond populations (BYN and PYO, respectively) split from HEL and LEV, 600 generations before present and experienced bottleneck for 300 subsequent generations. Each population consisted of 200 individuals. Like in the real stickleback genome, we simulated 20 chromosomes (sex chromosomes were not simulated). Each chromosome contained four LD blocks: one comprising 250 SNPs, and the other three 25 microsatellites each. The recombination rate was set to be $10^{-5}$ between loci within each of the LD blocks. In the three microsatellite LD blocks, mutation rates were specified to be $5 \times 10^{-6}$ (low mutation rate; Mi_l) $5 \times 10^{-5}$ (medium mutation rate; Mi_m) and $5 \times 10^{-4}$ (high mutation rate; Mi_h), with 10, 22 and 64 alleles, respectively. We then randomly picked 10 individuals from each of the four sub-populations, and used both SNPs and microsatellites as neutral markers to estimate $F_{ST}$. To simulate the quantitative traits, we randomly picked 20 individuals from each of the four sub-population as founders, and then used a full-sib design to simulate 10 full-sib families (five offspring per family) within each of the sub-populations which evolved through 30 generations using a simulation procedure similar to that used in Karhunen *et al.* [7].

This simulation scheme was used to model different divergence scenarios corresponding to varying intensity of selection. Specifically, different evolutionary forces over the 30 generations were applied as follows: in scenario (i), the populations bred completely randomly without any selection (neutral scenario; $Q_{ST} = F_{ST}$). In scenarios (ii) to (iv), populations were subjected to increasing levels of directional selection ($Q_{ST} > F_{ST}$) by multiplying the neutral pattern by a factor of 1.5, 2 or 4, respectively, thus generating weak (ii), moderate (iii) and strong selection (iv). All the genetic and environmental parameters needed in simulations were specified in the same way as in Karhunen *et al.* [7]. Due to the high computational cost of MCMC-based approach (approx. 24 h for each run), the whole simulation procedure was replicated 10 times, and the averaged performances of both Driftsel and QstFstComp on the replicated datasets were recorded.

# 3. Results

## 3.1. $F_{ST}$ comparisons: effects of marker type, genomic location and averaging approach

The estimated neutral baseline for the three SNP datasets based on the Weir & Cockerham [55] theta estimator of $F_{ST}$ was around 0.51, with very narrow credibility intervals (electronic supplementary

material, table S1). Hence, the mean and distribution of $F_{ST}$ values were not significantly affected whether estimated using all, non-genic or genic loci (electronic supplementary material, table S1; figure 1). However, all of these values were significantly higher than the $F_{ST} = 0.35$ estimated from the 12 microsatellite loci used by Karhunen *et al.* [8]. While the $F_{ST} = 0.37$ estimated from the *in silico* genotyped microsatellite loci was similar to that of Karhunen *et al.*'s [8] estimate, comparison of estimates calculated for the different sets of microsatellites differing in their variability (see electronic supplementary material, figure S4 for details of variability in these loci) revealed that the $F_{ST}$ estimates declined with increasing variability (electronic supplementary material, table S2; figure 1). The effect of marker variability on $F_{ST}$ was particularly clear from the distribution of $F_{ST}$ values: microsatellites with few alleles (1–4) were skewed towards extreme $F_{ST}$ values and looked most similar to equivalent distributions for SNPs (figure 1); those with intermediate numbers of alleles (5–8) had the broadest distribution of $F_{ST}$ and most strongly differed between the population pairs, whereas the ones with many alleles (9–21) were least diverged across the different population pairs (figure 1).

We further compared two approaches for combining $F_{ST}$ estimates across loci: average of ratios (AOR) and ratio of averages (ROA). Overall, ROA gave consistently higher estimates of $F_{ST}$ than AOR (figure 1). The relative difference between the two approaches differed between comparisons and data types, but was the smallest for the two least diverged populations, FIN-HEL and RUS-LEV. While the three SNP datasets produced identical $F_{ST}$ estimates under both averaging approaches, the relative order of marker categories in the microsatellites dataset changed drastically: the $F_{ST}$ estimates based on highly variable microsatellites were clearly the lowest when using ROA, whereas they were one of the highest when using AOR (figure 1*b*).

## 3.2. $F_{ST}$ comparisons: effects of variability-based ascertainment and minor allele filtering

We compared the $F_{ST}$ estimates from the full data with those estimated from loci ascertained using the two marine populations, FIN-HEL and RUS-LEV. The effects of ascertainment differed between population comparisons, datasets and averaging approaches (figure 1). Notably, the greatest impact among SNP-based estimates was in the comparison of populations used for the ascertainment, whereas the estimates for the two pond populations—neither included in the ascertainment—were virtually unchanged (figure 1*a*). Among microsatellites, variability-based ascertainment only affected estimates based on the least variable category (figure 1*b*).

Another approach to reduce the impact of rare, novel variants is to filter the data using minor allele frequency (MAF). We tested the MAF filtering using SNP data and found that it had clear effects on $F_{ST}$ estimates when using average of ratios (AOR) but only subtle effects when using ratio of averages (ROA; figure 2*a*). The effect on data ascertained by variability in marine populations was minimal (data not shown). This is consistent with the fact that the variability-based ascertainment itself heavily reduces the number of SNPs and the MAF filtering have only a minimal additional effect (figure 2*b*). Another notable effect of variability-based ascertainment was the changes in the shape of minor allele frequency: although the number of variable SNPs was reduced in all populations, the relative impact on numbers of low-frequency alleles was drastically different in the highly variable marine populations, especially on FIN-HEL, than in the small pond populations (figure 2*c*).

## 3.3. $F_{ST}$ comparisons: effects of recombination rate

The effects of local recombination rate on $F_{ST}$ between the two marine populations were statistically significant, but subtle. For SNPs, the average per-locus $F_{ST}$ was 0.1265 and 0.1352 (*t*-test: $p < 2.2 \times 10^{-16}$) in regions of low and high recombination rate, respectively, while for microsatellite loci with few alleles, they were 0.1403 and 0.1540 (*t*-test: $p < 0.05$; electronic supplementary material, figure S2c). The differences were consistent but not significant for microsatellite loci with an intermediate number of alleles (mean $F_{ST}$ 0.140 and 0.145, $p = 0.15$) while that for the most allele-rich loci were incongruent with the other categories of markers albeit not significantly so (0.1234 and 0.1186, $p = 0.36$; electronic supplementary material figure, S2c).

## 3.4. $F_{ST}$ comparisons: effects of outliers

Outlier analyses did not detect any outliers either in the SNP nor microsatellite datasets, suggesting that outliers had little influence on mean $F_{ST}$ estimates in our data (results not shown).

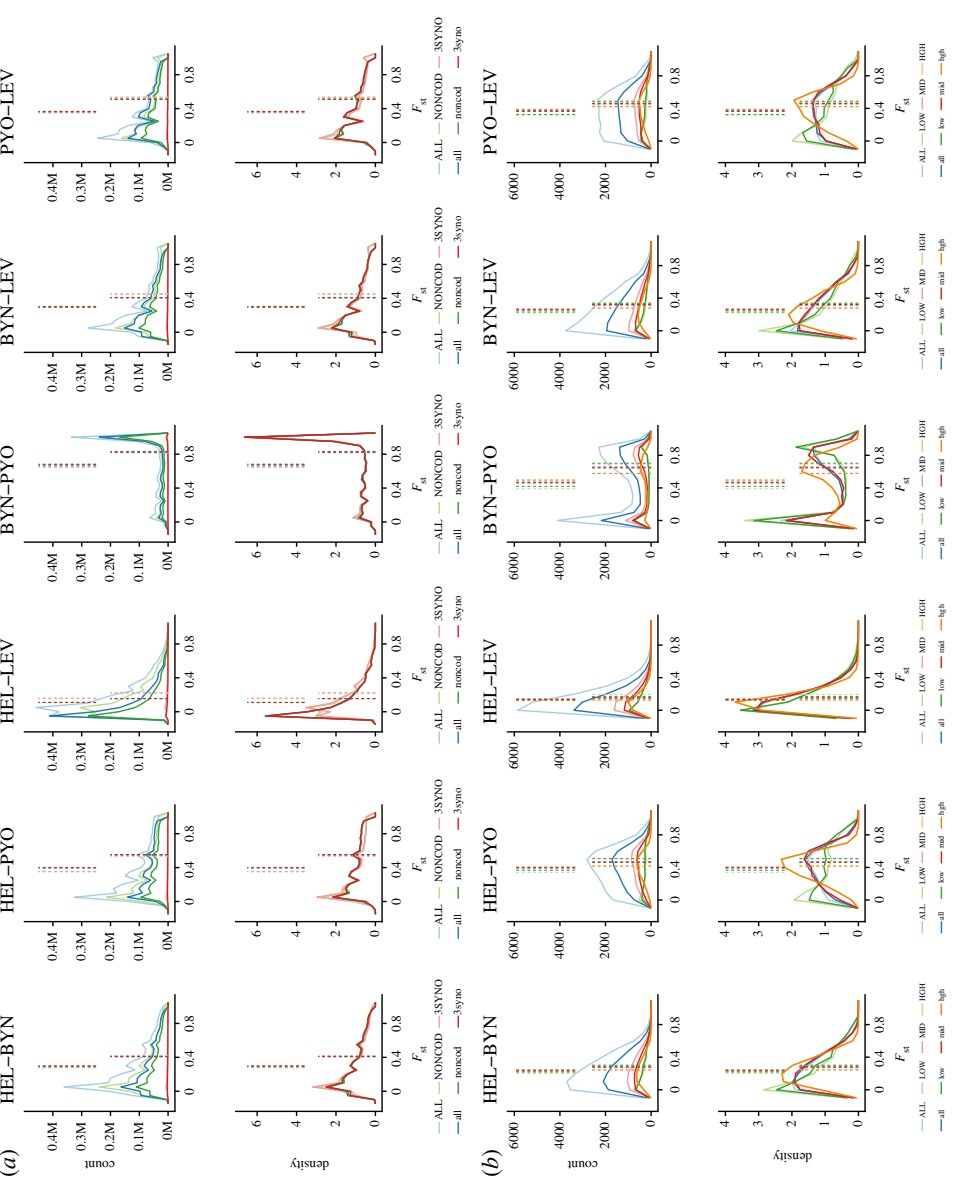

**Figure 1.** Distribution of per locus pairwise $F_{ST}$. Values of per locus $F_{ST}$ were estimated between all pairs of the four populations based on: (*a*) 3 806 181 SNP loci and (*b*) 18 824 microsatellite loci. Colours represent for each marker type the different datasets used to estimate $F_{ST}$ estimate: in (*a*) all SNP loci ('ALL' dataset; blue), non-coding/non-repeat sites ('NONCOD' dataset; green) and codon third position synonymous changes ('C3SYNO' dataset; red); light colours (in capitals) indicate full datasets whereas dark colours (in lowercase) show the distributions for loci ascertained by variability in marine populations. Absolute counts of sites (top) and density plots (bottom) are shown. Loci with minor allele frequency below 0.025 were discarded. In (*b*), $F_{ST}$ for full microsatellite data ('ALL' dataset; blue) and three subsets of data containing either low (1–4; 'LOW'), moderate (5–8; 'MID') or high (9–21; 'HIGH') number of alleles per locus. In all plots, dashed vertical lines indicate the mean value for the corresponding category: the upper and lower halves show the average of ratios and the ratio of averages, respectively.

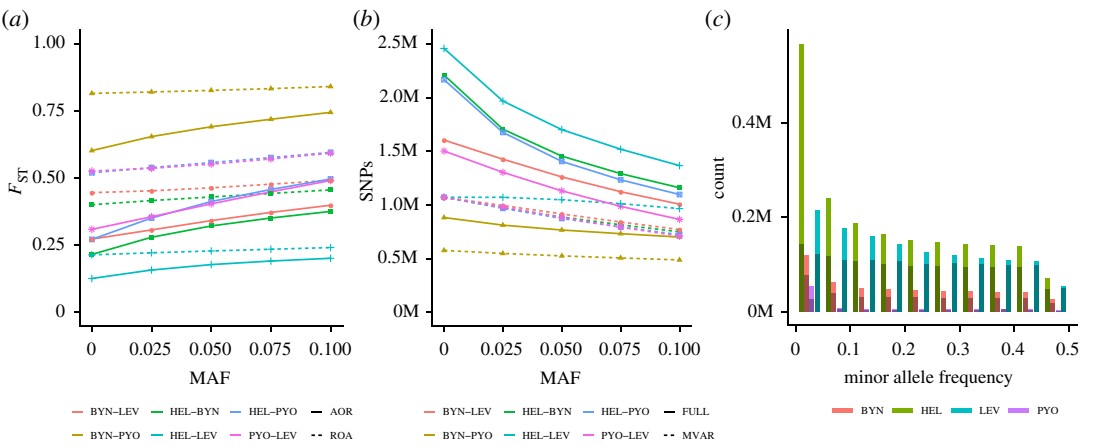

**Figure 2.** Effect of minor allele frequency (MAF) cut-offs on mean $F_{ST}$ estimated from the NONCOD SNP dataset. (a) Mean $F_{ST}$ for different population pairs as a function of MAF, (b) number of SNPs retained, (c) site frequency spectrum for the four populations using the bin width of 0.05. In (a), solid and dashed lines indicate the two averaging approaches, average of ratios (AOR) and ratio of averages (ROA) whereas in (b) they indicate full data (FULL) and loci ascertained by variability (MVAR). In (c), dark shading shows the effect of ascertainment by variability in marine populations.

## 3.5. $F_{ST}$ comparisons: effects of LD

For all three SNP datasets, pruning of SNPs with the distance thinning approach had little effect on the mean $F_{ST}$: it remained constant irrespective of the number of SNPs sampled for estimation (figure 3a–c). The LD-based pruning yielded quite different results: when pooled data were used, the mean $F_{ST}$ was heavily deflated for all three SNP datasets over much (0.1–0.7) of the parameter range (figure 3a–c). More specifically, this deflation was most marked at the lower end of this LD threshold range, generating a positive correlation between $F_{ST}$ and the proportion of SNPs sampled (figure 3a–c). Only when the LD-threshold was set to be less than 0.1 or greater than 0.7 (corresponding to selection of less than 2000 or greater than 20 000 SNPs in C3SYNO), the sampled unlinked SNPs yielded $F_{ST}$s similar to that in the full datasets (figure 3a–c). As we employed LD-based pruning separately for pond and marine populations, the $F_{ST}$ values were more similar (but still somewhat deflated) to those obtained with the thinning approach (figure 3a–c). Although the extent of LD differs in different populations (figure 3d), this is unlikely to explain the $F_{ST}$ deflation in the LD-based pruning.

## 3.6. $Q_{ST}$-$F_{ST}$ tests: effects of marker type and filtering

Irrespectively of the marker type or filtering criteria used, the Driftsel analyses detected significant signals of divergent natural selection for most of the traits while QstFstComp detected only very weak and non-significant signals of divergent selection (figure 4; electronic supplementary material, table S1–S7). In all analyses, Driftsel results were also fully consistent with earlier results based on 12 microsatellite loci ([8]; figure 4; electronic supplementary material, table S1, S5). Filtering of SNP data did not affect the outcomes: both Driftsel and QstFstComp yielded the same inference across datasets (i.e. ALL, NONCOD, C3SYNO; figure 4; electronic supplementary material, tables S1, S3, S5, S7). Marker type (i.e. microsatellites or SNP) did not influence the outcome of the Driftsel analyses: all traits displaying evidence for divergent natural selection with SNP markers did also with microsatellite markers (figure 4). Similarly, marker type did not influence the outcome of the QstFstComp analyses (figure 4; electronic supplementary material, table S4).

## 3.7. $Q_{ST}$-$F_{ST}$ tests: effects of marker ascertainment

Marker ascertainment did not influence the outcome of the QstFstComp analyses, regardless of the type of genomic dataset used (figure 4; electronic supplementary material, tables S3, S4, S7). The ascertainment did influence the outcome of the Driftsel analyses for microsatellite markers, however (figure 4). Specifically, Driftsel detected a significant signal of divergent selection on the multivariate behaviour phenotype ('AllB'; figure 4; electronic supplementary material, table S6) when *in silico* microsatellites were used. However, this effect disappeared when using only *in silico* microsatellites

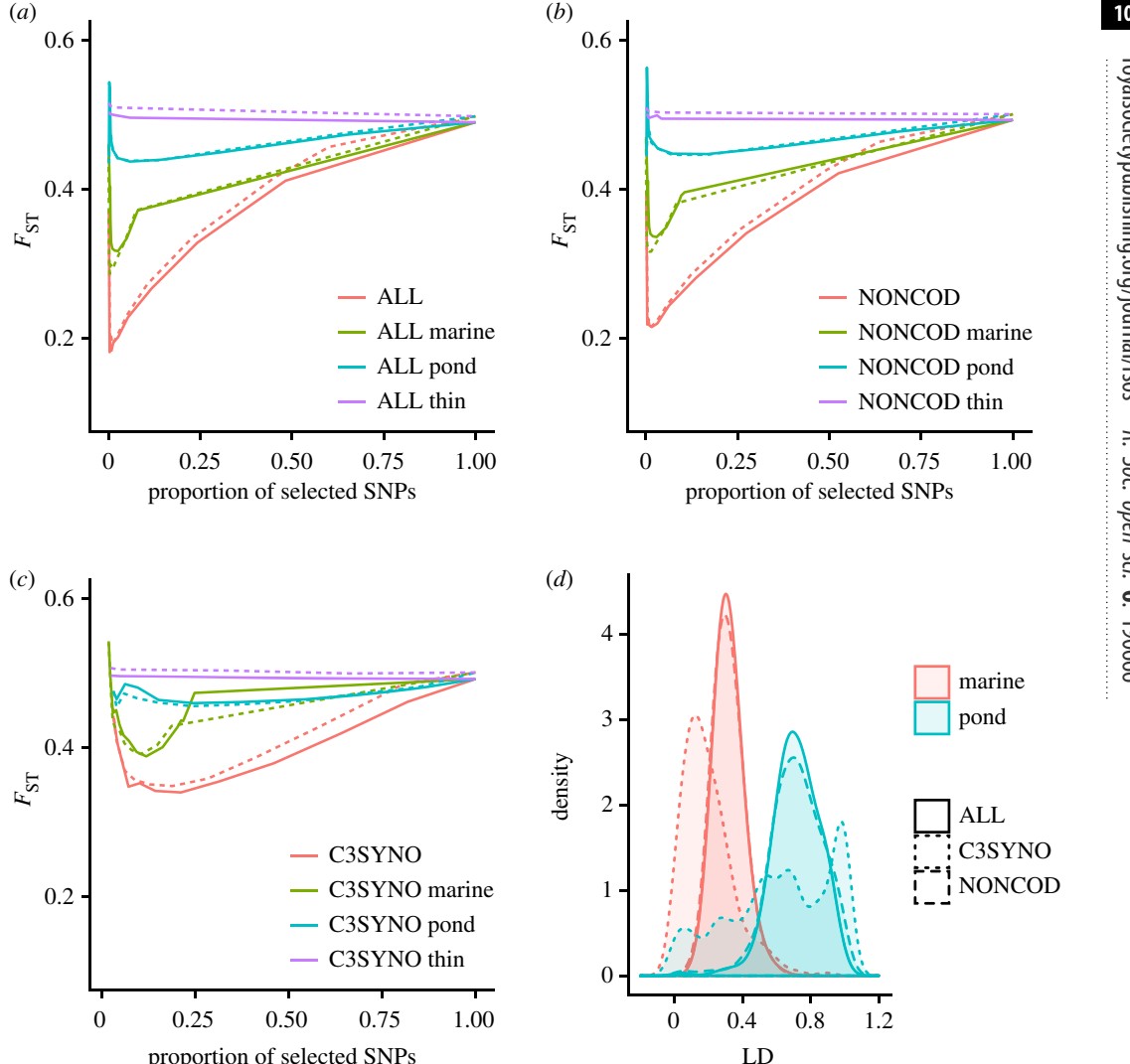

**Figure 3.** Effect of LD pruning on $F_{ST}$ estimated from different datasets. Results are shown for (*a*) all SNPs ('ALL' dataset), (*b*) non-coding/non-repeat sites ('NONCOD' dataset) and (*c*) codon third position synonymous changes ('C3SYNO' dataset). For each dataset, solid lines are colour coded to depict results obtained with the thinning approach (purple), or sliding window approach for all pooled populations (red), marine populations only (green) or pond populations only (blue). Dashed lines depict the same results after removing loci with MAF < 0.05. (*d*) The distribution of the average degree of linkage disequilibrium (LD) in marine and pond populations as estimated from non-overlapping 500 kb sliding windows across the genome.

with low to mid number of alleles (figure 4; electronic supplementary material, table S6). Moreover, the effect of marker ascertainment on the outcome of Driftsel was more pronounced when the H statistic [8] was computed (electronic supplementary material, figure S5). For this test, both NONCOD and C3SYNO ascertained SNP datasets failed to detect signal of divergent selection on behavioural traits ('AllB' and 'B1'; electronic supplementary material, figure S5) and morphological traits ('M2' and 'M3'; electronic supplementary material, figure S5). Similarly, the ALL SNP and Mi_h datasets failed to detect signals of selection for the first behavioural trait ('B1'; electronic supplementary material, figure S5) when using ascertained markers. The ALL SNP dataset also failed to detect the signal of selection for the second morphological trait ('M2'; electronic supplementary material, figure S5) when using ascertained markers.

## 3.8. Simulation study

$F_{ST}$ values estimated by the Driftsel method were 0.49, 0.47, 0.43 and 0.31 for the SNP, Mi_l, Mi_m and Mi_h datasets, respectively, thus approaching the $F_{ST}$ estimates obtained from the real stickleback data (electronic supplementary material, table S1 and S2). Under the scenarios (i) and

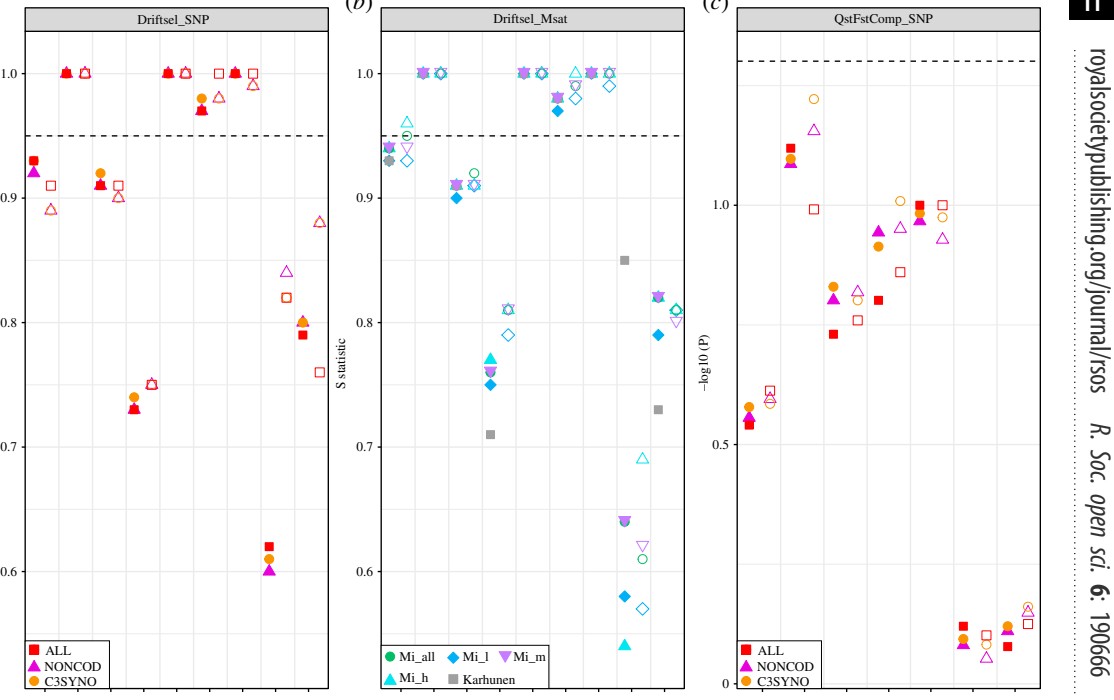

**Figure 4.** Effect of marker ascertainment, marker type and filtering of SNP dataset on $Q_{ST}$-$F_{ST}$ comparisons. Results of Driftsel analyses using SNP and microsatellite markers (*a,b*, respectively) and QstFstComp using SNP markers (*b*) are shown. (*b*) The Driftsel results with the 12 microsatellites used in [8] (Karhunen; grey squares) as well as for different sets of *in silico* microsatellites: all (Mi_all; green circles), ones with low number of alleles (Mi_l; blue diamonds); with intermediate number of alleles (Mi_m; purple triangles), and with high number of alleles (Mi_h; cyan triangles). (*a,c*) The results of Driftsel and QstFstComp, respectively, using all SNPs ('ALL' dataset, red squares); non-coding/non-repeat sites ('NONCOD' dataset, pink triangles) and codon 3rd position synonymous changes ('C3SYNO' dataset, orange circles). Values of the S statistic (Driftsel) or ($-\log10$) *p*-values (QstFstComp) are reported on the *y*-axis and dashed lines correspond to the significance threshold for each method. For all panels, filled and open shapes correspond to whether or not the full datasets (filled) or datasets with variability-based ascertained markers (open) were used.

(ii) (i.e. $Q_{ST} \approx 0.45$–0.68), the S statistics estimated from the Mi_h data were considerably higher than the estimates obtained with SNP or Mi_l datasets (electronic supplementary material, figure S6(a) and table S9), but in general, the averaged S statistics of all the datasets were under the significance threshold (0.95) to identify the divergent selection. In scenario (iii) (i.e. $Q_{ST} \approx 0.85$), the S statistic of Mi_h set started to go beyond the significance threshold, and had better power to identify the signature of selection compared with other datasets. In scenario (iv) with strong signature of selection (i.e. $Q_{ST} \approx 0.99$), the S statistic among all the datasets were consistent, and they all indicated strong selection signals (S $\approx 0.99$). By contrast, the *p*-values of all the datasets by using the QstFstComp method did not reach the significance level even in scenario (iv) (electronic supplementary material, figure S6(b)), and the power to detect signature of selection appeared to be consistently low (electronic supplementary material, table S9).

## 4. Discussion

In accordance with earlier suggestions [9,10], our results indicate that microsatellite markers can indeed yield an overly liberal neutral baseline for $Q_{ST}$-$F_{ST}$ tests, and this problem increases with increasing marker variability. Yet this made little difference for the outcome of the $Q_{ST}$-$F_{ST}$ tests in the stickleback data: SNP and microsatellite data yielded the same evolutionary inference. Similarly, we found little evidence to suggest that the genomic location of SNP-markers would markedly influence the $F_{ST}$ estimates. Filtering of the SNP data for minor alleles had some impact on the estimation of $F_{ST}$ when using AOR, but the magnitude of this impact was low and varied from one population

comparison to another. However, the ascertainment of the markers by variability in marine populations had a relatively large but inconsistent effect: it changed the $F_{ST}$ estimates when comparing populations used in ascertainment, but had virtually no impact on the comparison of pond populations. Worryingly, the direction of change differed between comparisons from slight increase to significant decrease. If the results are generalizable to other systems involving a greater number of populations with less extreme divergences, the choice of populations used for marker ascertainment may have a relatively large impact on $F_{ST}$ estimates, and thus, on the neutral baseline divergence estimation. Hence, our results highlight some challenges (and opportunities) which the earlier $Q_{ST}$-$F_{ST}$ studies based on low marker numbers did not meet. In the following, we will briefly discuss the implications of these findings for comparative studies of quantitative trait and molecular marker differentiation.

We compared two approaches for combining $F_{ST}$ estimates across loci: average of ratios (AOR) and ratio of averages (ROA). Although AOR has been shown to be affected by rare alleles and underestimate the $F_{ST}$ [29], it often seems the most natural way to compare subsets of genomic estimates (e.g. in sliding window analyses), and is probably still widely used [29]. Our analyses confirmed that ROA gives consistently higher estimates of $F_{ST}$ than AOR. The relative difference between the two approaches differed between comparisons, but, more interestingly, the relative order $F_{ST}$ estimates also changed within a comparison between different sets of markers: in our analysis of microsatellite data, the $F_{ST}$ estimates based on highly variable loci were clearly the lowest with ROA but among the highest with AOR. As a side note, when doing the analyses, we learned that the proper averaging approach, ROA, has been made unnecessarily difficult as many softwares only output estimates of the sitewise ratio of variances. For example, the popular software VCFtools could be easily changed to output also the numerator and denominator of the $F_{ST}$ estimates, and thus, allow computing ROA for arbitrary subsets of loci.

One of the major concerns in the application of $Q_{ST}$-$F_{ST}$ tests for adaptive differentiation has been the likely underestimation of neutral genetic differentiation when highly variable microsatellite markers are used [9,10]. Our results comparing $F_{ST}$ values estimated using a very large number of microsatellite loci with differing levels of variability do indeed suggest, in accordance with earlier studies [13,14], that highly variable loci deflate $F_{ST}$ estimates. However, to which degree this constitutes a concern for $Q_{ST}$-$F_{ST}$ studies depends also on the degree of differentiation in quantitative traits. In the case of the stickleback data analysed here, this made little difference for the outcome of the Driftsel analyses: although the $F_{ST}$ values were lower for microsatellite than for SNP loci, all the tests that rejected the null hypothesis of differentiation by drift with microsatellites also did so with SNPs. However, this point should be treated cautiously because of the nature of our data. Very high degree of quantitative trait differentiation among marine and freshwater nine-spined sticklebacks are well documented and have been attributed to divergent natural selection for a suite of ecologically important traits [8,37]. Such a marked differentiation is not the norm in empirical $Q_{ST}$-$F_{ST}$ studies where the degree of differentiation can be more subtle (e.g. [64,65]). Hence, the observed insensitivity of the $Q_{ST}$-$F_{ST}$ tests towards biases in the neutral baseline $F_{ST}$ should not be considered as a general rule. However, the particular case of the sticklebacks could be viewed as an exception confirming the rule: marker type and variability have the potential to bias the neutral baseline $F_{ST}$, and although they are less likely to affect inferences involving strong quantitative trait differentiation (i.e. high $Q_{ST}$), they should be particularly acknowledged when degree of quantitative trait differentiation is at low to moderate levels. This view is reinforced by the effect of marker ascertainment on the outcome of our $Q_{ST}$-$F_{ST}$ tests. Although the observed effect was trait-specific and confined to highly variable microsatellites in Driftsel analyses, we found that $Q_{ST}$-$F_{ST}$ inference can be influenced by marker type and filtering criteria, even when $Q_{ST}$ is high.

One critical assumption underlying $Q_{ST}$-$F_{ST}$ analyses is that the markers used for estimation of $F_{ST}$ are unlinked, as inclusion of tightly linked markers in $F_{ST}$ estimation can lead to biases in neutral expectations. This assumption is not discussed much in the $Q_{ST}$-$F_{ST}$ literature, where the major methodological concerns have instead been in the estimation of $Q_{ST}$ (e.g. [66,67]), as well as in the effect of marker variability on $F_{ST}$. One reason for the lack of concern over possible effects of LD is that most $Q_{ST}$-$F_{ST}$ studies have used relatively few markers, and the likelihood of sampling tightly linked loci with few markers is low. With genome-scale data, accounting for LD can become a major concern, as linked markers do not provide independent information about the evolutionary history and demography of the study populations [34,68]. Using two different approaches to account for LD, we discovered that the LD-based sliding window approach [46], which is a commonly used tool in genome-wide association and population genetic studies (e.g. [69–72]), may lead to gross underestimation of $F_{ST}$. The reason for this appeared to be the strong structure among our study populations that created an impression of LD when the data were pooled. The pond populations had

low nucleotide diversity (cf. figure 2c) and many loci showing variation in the marine populations were either fixed or nearly fixed in the ponds. Such differentiated loci showed high $F_{ST}$ between the populations but also appeared to be in high LD with each other and were preferentially pruned by the sliding window approach. As such, a severe downward bias in $F_{ST}$ ensued. In fact, LD pruning methods developed for populations with homogeneous LD structure are known to result in misleading outcomes when applied to data from structured populations [73]. Hence, care should be taken when pruning linked markers from multiple-population data, and as demonstrated here, a simple distance thinning approach can maintain the genetic structure of the original data and avoid downward bias in $F_{ST}$ which could lead to overly liberal $Q_{ST}$-$F_{ST}$ tests. Moreover, our results suggest that the LD-based approach may work relatively well when the LD is estimated from one population or a set of closely related populations; in our case, thinning based on pond populations worked the best (figure 3a–c).

The question of how loci with low minor allele frequencies should be dealt with in population genomic investigations is contentious. Because such loci exhibit high sampling variance, and carry very little information about populations' demographic histories, it is commonly argued that they should be excluded from $F_{ST}$ estimation (e.g. [31,33], but see [30]). We observed that $F_{ST}$ was sensitive to decisions about minor allele filtering cut-off points: the higher the cut-off point, the higher the $F_{ST}$ became. However, the magnitude of this effect varied from one population pair to another, apparently because of the fairly large differences in the frequency of minor alleles among populations. What may be useful to note is that the effect of minor allele filtering on $F_{ST}$ may depend on how it is calculated [29]. The mean $F_{ST}$s can be obtained by averaging locus-specific $F_{ST}$ estimates, or by first calculating the average (within and between populations) variances of all the SNPs and then estimating the mean $F_{ST}$ of those. Since the individual $F_{ST}$s of the low-frequency variants are small, they weigh down the average $F_{ST}$ as obtained with the first approach. However, since the low-frequency variants have very small allelic variances, their influence on the mean $F_{ST}$ as estimated from the second approach (which is the one used by Driftsel and other $Q_{ST}$-$F_{ST}$ comparison methods) is small. Hence, given their relatively small contribution to $F_{ST}$, and in order to err on the conservative side, we believe that it would be justified to remove low-frequency alleles (e.g. MAF < 0.025) when estimating the $F_{ST}$ baseline for $Q_{ST}$-$F_{ST}$ tests. Their inclusion would have only a minor impact and tend to make tests of adaptive hypotheses slightly more liberal.

We found that $F_{ST}$ estimates for SNPs and less variable microsatellite loci were on average slightly, but significantly higher in regions of high than low recombination. While this effect was small, it is noteworthy since the difference was in the opposite direction than expected. Namely, background and hitchhiking selection are expected to have stronger effects on allele frequencies in areas of low than high recombination. This is because linkage between neutral and selected loci is tighter in the areas of low recombination [27,28,74]. In fact, several studies have documented negative correlations between population differentiation and recombination rate [75–82]. On the other hand, introgression between fish species was recently shown to be more common in regions of high recombination [83] and possibly the weaker linkage to negatively selected sites allow loci with smaller adaptive effects to diverge between populations. This would create correlation between genetic divergence and recombination rate, and could explain the slightly elevated $F_{ST}$ values. Although the effect we observed was so subtle that it did not have practical consequences for our $Q_{ST}$-$F_{ST}$ comparisons, recombination rate and degree of population differentiation are often found to go hand-in-hand. Future $Q_{ST}$-$F_{ST}$ studies should thus pay particular attention to how the markers in test panels are distributed in respect to variation in recombination rate or its proxies, such as their location relative to centromeres (e.g. [82]).

Interestingly, we did not detect any outlier loci in our analyses, suggesting that their impact on baseline $F_{ST}$ estimates, and hence on the outcome of our $Q_{ST}$-$F_{ST}$ tests, can be assumed to be negligible. However, the fact that no outliers were detected does not mean that they did not exist in the data: given the extremely strong impact of genetic drift on the genetic constitution of the pond populations, as for instance reflected in the generally very high $F_{ST}$ estimates and lack of low-frequency alternative alleles, the outlier tests were probably ill-suited for detecting footprints of selection against such a highly divergent neutral background (e.g. [36]). However, even if some outliers might have become included in the data, it is likely that their impact is small given that a few high-$F_{ST}$ loci make little difference for the overall $F_{ST}$ when averaged over a large number of unlinked and presumably neutral loci.

$Q_{ST}$-$F_{ST}$ comparisons have a long history in evolutionary biology (reviewed in [4]), and recent years have seen several methodological refinements to the basic approach (e.g. [8,56–58,84]). Here, we employed two recently developed approaches and although our aim was not to provide formal

performance comparisons between the two methods, we discovered that the Driftsel approach recovered consistently more evidence for selection in our data than the QstFstComp approach. This was expected, as the Driftsel approach has been shown to have more power to detect footprints of selection than other types of $Q_{ST}$-$F_{ST}$ tests, especially when the number of populations is small and the impact of random genetic drift is strong [58]. Our simulations further confirm the validity of this inference. Under a scenario of profound quantitative trait divergence ($Q_{ST} > F_{ST}$), Driftsel was consistently able to detect the signal of selection, irrespective of the marker set used, whereas QstFstComp approach failed to do so. Under the scenario of intermediate differentiation, neither of the two approaches was able to pick up the signal of selection, albeit the Driftsel approach applied to highly variable markers showed tendency towards this. This observation underlines the validity of the concerns associated with the use of highly variable markers in $Q_{ST}$-$F_{ST}$ comparisons [9,10]. The generally poor performance of both Driftsel and QstFstComp approaches in detecting signatures of selection in our simulated datasets can be explained by two factors. First is the high level of neutral baseline differentiation used in the simulations, and the second is the small number of populations used. As shown earlier, the performance of both approaches increases with decreasing impact of drift and increasing number of populations included into analyses [58].

Another illuminating lesson from our comparisons was that increasing the number of microsatellite loci (from 12 to 2000) used for $F_{ST}$ estimation did not influence the outcome of our $Q_{ST}$-$F_{ST}$ tests. However, given the many possible sources of error and bias in estimating neutral baseline differentiation, including as many judiciously filtered markers as possible, may be warranted. This also highlights the need for further developments in the current $Q_{ST}$-$F_{ST}$ testing tools. Although Driftsel is in principle capable of handling infinite number of loci, the run-times for individual tests quickly become prohibitive once the number of markers starts to exceed a few thousand. A nature of the stochastic sample-based MCMC algorithm is that the larger number of SNPs included, the slower the algorithm will converge to the target posterior distribution, and more iterations will be needed. Hence, fine-tuning of the underlying algorithm to meet the demands of modern-day marker-panels is needed. For example, a deterministic variational Bayes (VB) algorithm [85,86] might provide a faster alternative to analyse large-scale SNP data with the Driftsel model.

Finally, it may be also worth stressing that $Q_{ST}$-$F_{ST}$ tests are prone to biases stemming from the choice of study design used to obtain quantitative genetic parameters needed for the estimation of $Q_{ST}$. Specifically, the use of full-sib design, such as used in the present study, yields estimates of additive genetic variance potentially confounded with dominance variance [87]. However, dominance variance tends to reduce, rather than inflate $Q_{ST}$ [88,89]. Thus, even if non-additive inheritance were at play in the differentiation among our stickleback populations, our results would most likely still be rather conservative in respect to the detection of signals of divergent selection.

# 5. Conclusion

In conclusion, our empirical investigation of genetic variation and differentiation based on large numbers of SNP and microsatellite loci shows that both under- and overestimation of the neutral baseline level of differentiation can occur depending on the filtering and ascertainment of marker data. The results highlight the fact that marker ascertainment, but sometimes also minor allele filtering, might influence the outcome of $Q_{ST}$-$F_{ST}$ tests through their effects of $F_{ST}$, especially if the degree of quantitative trait divergence is low. Similarly, while pruning of tightly linked markers from the SNP panels is needed to avoid biased neutral estimates of differentiation, standard filtering methods developed for data from unstructured populations may result in serious biases if populations are strongly differentiated. However, once founded on carefully assembled SNP-panels and sound quantitative genetic data, $Q_{ST}$-$F_{ST}$ comparisons continue to provide a useful framework for testing the adaptive basis of population differentiation in ecologically important quantitative traits. Our findings and considerations suggest that for $Q_{ST}$-$F_{ST}$ comparisons, SNP-markers are preferable over microsatellites, and if microsatellites are to be used, loci from the lower end of the allele number distribution should be preferred over those from the higher end.

Data accessibility. R scripts for the simulation study are attached as electronic supplementary material. SNP and *in silico* microsatellite data are available within the Dryad Digital Repository: https://dx.doi.org/10.5061/dryad.c2fqz6140 [90].
Authors' contributions. J.M. conceived the study; A.L. and Z.L. designed the methodology; all authors contributed to collecting of the data; A.L. processed the genomic data; A.L. and Z.L. analysed the data; J.M. and A.F. led the writing of the manuscript. All authors contributed critically to the drafts and gave final approval for publication.

Competing interests. We have no competing interests.

Funding. We were supported by the Academy of Finland (grant nos 134728, 250435 and 265211 to J.M.).

Acknowledgements. We thank M.D. Edge and an anonymous reviewer for their comments that helped us to improve the manuscript. We thank J. De Faveri for comments and linguistic check (remaining errors introduced by us) of an earlier version of this manuscript. Thanks are also for all the people who helped in obtaining the data. Special thanks to S. Varadharajan, L. Nederbragt and K. Jacobsen for their generosity in allowing us tap into the unpublished nine-spined stickleback reference genome. We are also grateful to P. Momigliano and B. Guo for discussions relevant to $F_{ST}$ outlier detection. Computational analyses were enabled by the resources of CSC—IT Center for Science, Finland.

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
