## [Reviewer comments · Royal Society Open Science]

Review History

RSOS-190666.R0 (Original submission)

Review form: Reviewer 1

Is the manuscript scientifically sound in its present form?

No

Are the interpretations and conclusions justified by the results?

No

Is the language acceptable?

Yes

Is it clear how to access all supporting data?

No

Do you have any ethical concerns with this paper?

No

Have you any concerns about statistical analyses in this paper?

No

Recommendation?

Reject

Comments to the Author(s)

Before I begin, let me state up front that I have reviewed this paper in a previous incarnation for a different journal. That being said, I re-read the paper submitted here anew.

I believe that the part of this paper that confirms that the previously published stickleback results with microsatellite hold up with the more reliable markers is valid and should be published somewhere.

Having said that, I believe the more general statements made in this paper have little support from the new work reported here. It is not possible to use empirical results to test for bias in methods—for the simple reason that we do not objectively know the truth to compare to. The simulation results are too limited to justify many general conclusions. As a result, what we gain from this paper is fairly slim.

I believe that in a number of places the paper over-interprets its findings. I'll try to flag specific instances in my comments below.

Specific comments (by authors' original line number):

10-11: "both minor alleles and linkage disequilibrium pruning influence FST...that could render Qst FST comparisons overly conservative." No information is given anywhere in the paper to support that claim that minor allele filters caused the methods to be "overly conservative". Yes proper filtering changed the answer, but it has not been shown that the result is not more accurate (as indeed all other work on this would suggest is the case).

12: "comparisons can be insensitive to the choice of marker type" - I do not believe this is true. While it is certainly possible (and even common) that a biased procedure can return the same binary answer as a less-biased procedure to a hypothesis test, this does not mean that the method is insensitive to the bias. QST FST comparisons based on microsatellites will give biased answers; the fact that the effect is so large in this particular case such that that (rather large) bias does not change the conclusion to reject a null hypothesis is no particularly informative in any general context. Similarly in line 365, I would argue that the fact that sometimes QST is much larger than (true) FST (such as is the case in this specific example) does not mean that the biases caused by imperfect markers are not a concern.

102: Why was a reduced set of populations used?

185: Specify here what averaging method was used to combine information about FST across loci. Later (line 411) it becomes clear that a biased form of averaging was used. It is well known that averaging the point estimates of FST of multiple loci gives a biased estimate of the FST produced by the underlying biological process. Weir and Cockerham showed that the method of taking the ratio of the average numerator and average denominator for FST across loci is much less biased. Results based on an estimation method already known to be flawed won't be helpful.

Figure 2 b+c are not necessary.

284: What does it mean that the “increase was consistent with the proportion of sites discarded”? As far as I know there is no theoretical prediction on this to say that the effect was consistent.

298: “neither” □ “either”

375: I appreciate that the authors are saying that their results for sticklebacks should not be considered a general rule. But this is a bit of a strawman – the results from this one case really don’t inform us about the general pattern, and that is the whole context of the paper.

404: I believe the authors mis-cite reference 72. This paper is about ascertainment bias, not about minor allele frequency filtering. There is truly no real controversy about whether lower MAF loci give biased estimates of FST.

464: It is not surprising that increasing the number of loci for FST -estimation did not help, because the main source of sampling error in most QST FST comparisons is uncertainty in QST, not FST.

A paper that perhaps ought to be cited is Wang’s paper on correcting bias for FST with microsatellites <https://www.ncbi.nlm.nih.gov/pubmed/25891752>.

Review form: Reviewer 2 (Michael Edge)

Is the manuscript scientifically sound in its present form?

Yes

Are the interpretations and conclusions justified by the results?

Yes

Is the language acceptable?

Yes

Is it clear how to access all supporting data?

Yes

Do you have any ethical concerns with this paper?

No

Have you any concerns about statistical analyses in this paper?

Yes

Recommendation?

Accept with minor revision (please list in comments)

Comments to the Author(s)

PDF attached (Appendix A).

Decision letter (RSOS-190666.R0)

30-May-2019

Dear Dr Fraimout,

The editors assigned to your paper ("Effects of marker type and filtering criteria on QST-FST comparisons") have now received comments from reviewers.

Both reviewers raise some substantive comments that will need addressing. As the Associate Editor comments, in particular you will need to focus on the interpretation of data and conclusions, which are often too strong. We would like you to revise your paper in accordance with the referee and Associate Editor suggestions which can be found below (not including confidential reports to the Editor). Please note this decision does not guarantee eventual acceptance and your revised manuscript may be re-reviewed.

Please submit a copy of your revised paper before 22-Jun-2019. Please note that the revision deadline will expire at 00.00am on this date. If we do not hear from you within this time then it will be assumed that the paper has been withdrawn. In exceptional circumstances, extensions may be possible if agreed with the Editorial Office in advance. We do not allow multiple rounds of revision so we urge you to make every effort to fully address all of the comments at this stage. If deemed necessary by the Editors, your manuscript will be sent back to one or more of the original reviewers for assessment. If the original reviewers are not available, we may invite new reviewers.

- Data accessibility

It is a condition of publication that all supporting data are made available either as supplementary information or preferably in a suitable permanent repository. The data accessibility section should state where the article's supporting data can be accessed. This section should also include details, where possible of where to access other relevant research materials such as statistical tools, protocols, software etc can be accessed. If the data have been deposited in an external repository this section should list the database, accession number and link to the DOI

for all data from the article that have been made publicly available. Data sets that have been deposited in an external repository and have a DOI should also be appropriately cited in the manuscript and included in the reference list.

If you wish to submit your supporting data or code to Dryad (<http://datadryad.org/>), or modify your current submission to dryad, please use the following link:
<http://datadryad.org/submit?journalID=RSOS&manu=RSOS-190666>

- **Competing interests**

- **Authors' contributions**

- **Acknowledgements**

- **Funding statement**

Kind regards,

Alice Power

on behalf of Professor Peter Visscher (Associate Editor) and Steve Brown (Subject Editor)

Associate Editor's comments (Professor Peter Visscher):

Please revise the manuscript taking the multiple comments from reviewers #1 and #2 into account. Both reviewers comment that at times the findings are over-interpreted or the

conclusions too strong. Depending on your response to the reviewers and the revised manuscript, the Editors will decide whether to send the revision out for re-review.

Comments to Author:

Reviewers' Comments to Author:

Reviewer: 1

Comments to the Author(s)

Before I begin, let me state up front that I have reviewed this paper in a previous incarnation for a different journal. That being said, I re-read the paper submitted here anew.

I believe that the part of this paper that confirms that the previously published stickleback results with microsatellite hold up with the more reliable markers is valid and should be published somewhere.

Having said that, I believe the more general statements made in this paper have little support from the new work reported here. It is not possible to use empirical results to test for bias in methods—for the simple reason that we do not objectively know the truth to compare to. The simulation results are too limited to justify many general conclusions. As a result, what we gain from this paper is fairly slim.

I believe that in a number of places the paper over-interprets its findings. I'll try to flag specific instances in my comments below.

Specific comments (by authors' original line number):

10-11: "both minor alleles and linkage disequilibrium pruning influence FST...that could render Qst FST comparisons overly conservative." No information is given anywhere in the paper to support that claim that minor allele filters caused the methods to be "overly conservative". Yes proper filtering changed the answer, but it has not been shown that the result is not more accurate (as indeed all other work on this would suggest is the case).

12: "comparisons can be insensitive to the choice of marker type" - I do not believe this is true. While it is certainly possible (and even common) that a biased procedure can return the same binary answer as a less-biased procedure to a hypothesis test, this does not mean that the method is insensitive to the bias. QST FST comparisons based on microsatellites will give biased answers; the fact that the effect is so large in this particular case such that that (rather large) bias does not change the conclusion to reject a null hypothesis is no particularly informative in any general context. Similarly in line 365, I would argue that the fact that sometimes QST is much larger than (true) FST (such as is the case in this specific example) does not mean that the biases caused by imperfect markers are not a concern.

102: Why was a reduced set of populations used?

185: Specify here what averaging method was used to combine information about FST across loci. Later (line 411) it becomes clear that a biased form of averaging was used. It is well known that averaging the point estimates of FST of multiple loci gives a biased estimate of the FST produced by the underlying biological process. Weir and Cockerham showed that the method of taking the ratio of the average numerator and average denominator for FST across loci is much less biased. Results based on an estimation method already known to be flawed won't be helpful.

Figure 2 b+c are not necessary.

284: What does it mean that the “increase was consistent with the proportion of sites discarded”? As far as I know there is no theoretical prediction on this to say that the effect was consistent.

298: “neither” □ “either”

375: I appreciate that the authors are saying that their results for sticklebacks should not be considered a general rule. But this is a bit of a strawman – the results from this one case really don’t inform us about the general pattern, and that is the whole context of the paper.

404: I believe the authors mis-cite reference 72. This paper is about ascertainment bias, not about minor allele frequency filtering. There is truly no real controversy about whether lower MAF loci give biased estimates of FST.

464: It is not surprising that increasing the number of loci for FST -estimation did not help, because the main source of sampling error in most QST FST comparisons is uncertainty in QST, not FST.

A paper that perhaps ought to be cited is Wang’s paper on correcting bias for FST with microsatellites <https://www.ncbi.nlm.nih.gov/pubmed/25891752>.

Reviewer: 2

Comments to the Author(s)
PDF attached.

Author's Response to Decision Letter for (RSOS-190666.R0)

See Appendix B.

Decision letter (RSOS-190666.R1)

16-Sep-2019

Dear Dr Fraimout,

I am pleased to inform you that your manuscript entitled "Effects of marker type and filtering criteria on QST-FST comparisons" is now accepted for publication in Royal Society Open Science.

You can expect to receive a proof of your article in the near future. Please contact the editorial office (openscience_proofs@royalsociety.org and openscience@royalsociety.org) to let us know if

you are likely to be away from e-mail contact -- if you are going to be away, please nominate a co-author (if available) to manage the proofing process, and ensure they are copied into your email to the journal.

Kind regards,

Lianne Parkhouse
Royal Society Open Science
openscience@royalsociety.org

on behalf of Professor Peter Visscher (Associate Editor) and Steve Brown (Subject Editor)
openscience@royalsociety.org

Appendix A

Review of Li et al., “Effects of marker type and filtering criteria on QST-FST comparisons”

General Comments

This manuscript takes up an interesting question about QST-FST comparisons used to test for selection on quantitative traits. In particular, the FST estimate often depends on aspects of genetic marker type and marker selection. (Method of estimation matters too but is not considered in the manuscript; all analyses focus on Weir-Cockerham estimation, which is the consensus choice.) The effects of marker type and marker selection on FST have been considered in many contexts, but the authors note that it has received less attention in the QST-FST literature. Overall, I felt that the claims made in the manuscript were justified by the data presented, and the scripts used for data analysis and generation of simulation data have been made available. Most of the below is presented in the spirit of showing the authors my constructive reactions to the work, and I do not think that they all need responses in order to meet the objective peer review criteria of *Royal Society Open* as I understand them.

The authors pursue questions about FST estimation primarily in a dataset of several stickleback populations and also in a limited simulation study inspired by stickleback demography. (I felt that the simulation arm of the study was not used as fully as it might have been and also that it could have been larger to give clearer resolution on the results.) In general the authors report changes to estimated FST values that are sensible and expected given prior literature, but these variations do not cause much difference in the reported QST-FST comparisons. Presumably the simulation framework could be used to find parameter regimes where the choices matter more.

One issue that puzzled me while reading the manuscript was not the fault of the authors but arises from the long history of FST in population genetics. As the authors know, FST has been variously interpreted as (i) a parameter of a data-generating process (e.g. of an evolutionary model of two or more subdivided populations) or as a function of those parameters and (ii) as a summary statistic describing genetic differentiation (e.g. as a proportion of variance). If one views FST as a parameter, then standard FST estimators for microsatellites (e.g. GST-style estimators) are often viewed as biased downward (e.g. Whitlock, 2011). If, on the other hand, one views FST as a descriptive statistic, then it is generally acknowledged to be smaller for more variable markers (e.g. Holsinger & Weir, 2009). QST-FST comparisons can be viewed from either point of view. The $QST \approx FST$ result has been derived many times independently, but in the derivations with which I am most familiar, FST is generally viewed either as (i) a function of parameters that is estimated accurately by low-mutation-rate markers (e.g. Whitlock 1999; Koch 2019 *Genetics*) or (ii) a statistic describing neutral differentiation specifically at biallelic markers (usually because the loci affecting the trait are taken to be biallelic; e.g. Berg & Coop 2014; Edge & Rosenberg 2015 *Human Biology*). I don't think that using a Weir-Cockerham FST computed from microsatellites makes sense under either view---under view (i) it is biased downward as an estimate of the parameter of genuine interest, and under view (ii) it is a description of differentiation at the wrong class of markers, which is unlikely to match differentiation at the right class of markers. It is possible that an alternative estimator developed for microsatellites, such as Slatkin's (1995) RST, could be a good estimator of the parameter of interest under view (i), as long as its assumptions accurately reflect the markers' history (e.g. for RST, the stepwise mutation model would have to be a reasonable fit). In some ways these issues are orthogonal to the manuscript, which asks whether using standard FST estimators affects the QST-FST comparison empirically. The answer seems to be that it easily could do so in some cases but doesn't for the specific empirical data in hand. That said, I think that some discussion of these

points would clarify the statistical framework that's implicit in QST-FST comparisons, and I do think that comparison with other microsatellite-based estimators such as RST would enrich the story here.

Another discussion point I would have liked to see was more talk about marker ascertainment. In systems with less fully developed genomic resources than sticklebacks, marker ascertainment may lead to an overrepresentation of highly heterozygous markers. There are already data on this in the manuscript, but I didn't recall comments on this specific view of the issue.

Specific Comments

"However, since per nucleotide mutation rates are highly variable, and may vary depending on their genomic location [20, 21], SNP loci may not automatically yield less biased neutral baseline estimates of divergence than microsatellites"

Biased as an estimator of what in this case?

"this would be indicative of uniform stabilizing selection that has prevented populations from diverging less than would be expected under random genetic drift alone"

The phrasing is confusing. Perhaps changing "less than" to "as much as" would help.

"Using the inferred rate, the SNP dataset NONCOD and the microsatellite data were split into subsets of loci with low (1-7 cM/Mb) and high (7-15 cM/Mb) local recombination rates, discarding the loci with either no rate estimate or an estimate from the extreme ends of the rate distribution (Fig. S3a). The microsatellite loci were further divided to those with few (1-4), intermediate (5-8) and many (9- 21) alleles (Fig. S3b), and FST was separately estimated for the different subsets"

-Is there a justification for these specific bins (e.g. are these median or tertile splits)? Any justification for these bins?

-I would be more curious to see results for binning msats by expected heterozygosity rather than by numbers alleles. As shown in the supplement, these are correlated, but heterozygosity has a direct link to FST.

-Are these the total numbers of alleles or the number of non-reference alleles? (And if total, why are there some msats with 1 allele?)

It would help the reader to give a few sentences more about the differences between the driftsel model and standard QST-FST comparisons. In particular, the sentence "it can detect selection even when QST = FST" is confusing without further details given the rationale for QST-FST comparisons explained in the introduction.

"Simulations were based on the most likely demographic history of real nine-spined stickleback populations"

In what sense is this demographic history "most likely"?

"Each chromosome contained four LD blocks: one comprising of 250 SNPs, and the other three of 25 microsatellites each"

Why are all SNPs on a chromosome in LD but not in LD w/ the msats?

"In scenario (ii) to (iv), populations were subjected to increasing levels of directional selection ($QST > FST$) by multiplying the neutral pattern by a factor of 1.5, 2 or 4, respectively, thus generating weak (ii), moderate (iii) and strong selection (iv)."

-What does “multiplying the neutral pattern” mean?

“The whole simulation procedure was replicated 10 times, and the averaged performances of both Driftsel and QstFstComp on the replicated data sets were recorded.”

-Would it be possible to increase the number of simulations? 1000 would be best but might be inaccessible—if so, I would suggest giving the reader a sense of how long these simulations take. At least 100 would give a lot more precision for these results.

“our results indicate that microsatellite markers are indeed likely to yield an overly liberal neutral baseline for QST-FST tests,”

I think this sentence is stated a little too strongly given that the authors have not identified cases where it does matter.

I found myself somewhat confused by the explanation of the pattern of FST results produced by the LD thinning. The explanation seems to turn mostly on both (a) the very different within-population LD in different populations and (b) the pruning of sites with fixed or near-fixed differences in the pooled data. I am missing what (a) has to do with (b): the fixed or nearly-fixed sites presumably should have little LD w/ neighboring markers within populations, so I don't see how the different levels of within-population LD matter. It seems to me that if you have fixed differences, then those will tend to show up as being in high LD (across populations) with other fixed differences, so if there are many fixed differences in a region, then many will be pruned on the basis of pooled data. If pruning affects these markers more strongly than it does markers that are less differentiated, then FST might be biased downward.

This review was written by Michael D. Edge.

Appendix B

Associate Editor's comments (Professor Peter Visscher):

Please revise the manuscript taking the multiple comments from reviewers #1 and #2 into account. Both reviewers comment that at times the findings are over-interpreted and the conclusions too strong. Depending on your response to the reviewers and the revised manuscript, the Editors will decide whether to send the revision out for re-review.

Please see our detailed responses below. We have now tuned down the claims and conclusions, which were perceived to over-interpret the results. Moreover, we have expanded our analyses following both reviewers' comments. We now provide results on the effect of averaging approach in the estimation of F_{ST} and marker ascertainment on the outcome of Q_{ST} - F_{ST} comparisons. These new results strengthen our conclusions and allow us to better discuss the effects of marker type and filtering criteria on Q_{ST} - F_{ST} comparisons.

Reviewers' Comments to Author:

Reviewer1:

Before I begin, let me state up front that I have reviewed this paper in a previous incarnation for a different journal. That being said, I re-read the paper submitted here anew.

Thank you for sharing this information and to give the manuscript another careful read.

I believe that the part of this paper that confirms that the previously published stickleback results with microsatellite hold up with the more reliable markers is valid and should be published somewhere. Having said that, I believe the more general statements made in this paper have little support from the new work reported here. It is not possible to use empirical results to test for bias in methods—for the simple reason that we do not objectively know the truth to compare to.

Our intention here was to provide an empirical test of the earlier proposals (cf. Edelaar & Björlund 2011, Edelaar et al. 2011) that Q_{ST} - F_{ST} approaches based on microsatellite markers can be overly liberal in favoring selective explanations as compared to ones based on SNPs. Since this has never been done before, to the best of our knowledge, it seems obvious that empirical approach can be used to test the a priori prediction that the choice of markers can cause bias. While it is true that firm answer cannot be obtained, an empirical test is possible. This how science works: we test predictions. Nevertheless, we have now revised the appropriate sections of the manuscript (see detailed responses below) to make sure that we do not overstate our case.

The simulation results are too limited to justify many general conclusions. As a result, what we gain from this paper is fairly slim. I believe that in a number of places the paper over-interprets its findings. I'll try to flag specific instances in my comments below.

We acknowledge that our simulation study could be improved by adding more scenarios. We now make this point clearer in the manuscript. Unfortunately, a comprehensive simulation study would require a substantial increase in both number of scenarios and simulation runs, which would be

computationally expensive and even prohibitive (see also response to Reviewer 2). As said above, we have carefully checked and edited the manuscript to avoid over-interpretations. Please note also that after this revision (in response to comments by reviewer #2), we have also evaluated the effect of ascertainment bias on F_{ST} estimates and Q_{ST} - F_{ST} tests. Hence, there is more meat on bone in the revised version.

Specific comments (by authors' original line number):

10-11: "both minor alleles and linkage disequilibrium pruning influence F_{ST} ...that could render Q_{ST} F_{ST} comparisons overly conservative." No information is given anywhere in the paper to support that claim that minor allele filters caused the methods to be "overly conservative". Yes proper filtering changed the answer, but it has not been shown that the result is not more accurate (as indeed all other work on this would suggest is the case).

We have now removed the part of the sentence suggesting overly conservative results in the abstract, and kept only what can be concluded from the results: both minor alleles and linkage disequilibrium pruning influenced F_{ST} estimations. See also the detailed responses below and L340-360 in the main text, as additional analyses now show that the effect of MAF filtering also depends on the type of F_{ST} -averaging method used.

12: "comparisons can be insensitive to the choice of marker type" – I do not believe this is true. While it is certainly possible (and even common) that a biased procedure can return the same binary answer as a less-biased procedure to a hypothesis test, this does not mean that the method is insensitive to the bias. Q_{ST} F_{ST} comparisons based on microsatellites will give biased answers; the fact that the effect is so large in this particular case such that that (rather large) bias does not change the conclusion to reject a null hypothesis is not particularly informative in any general context.

We agree with the reviewer that this is not to say that the method is insensitive to bias, but that the particular outcome of this study was. Here, the Q_{ST} - F_{ST} comparisons were insensitive to the choice of marker type, and arguably, the same could be true in other cases as well. We have checked this passage now and make it clear that this is specific to cases where Q_{ST} is high, such as in the present stickleback case.

Similarly, in line 365, I would argue that the fact that sometimes Q_{ST} is much larger than (true) F_{ST} (such as is the case in this specific example) does not mean that the biases caused by imperfect markers are not a concern.

Yes, we acknowledge that such bias are of concern always, although with possibly less marked consequences for "high Q_{ST} - low F_{ST} " situations. Generally, one could argue that imperfect markers are a general concern, even outside the scope of Q_{ST} - F_{ST} comparisons, but we do not deem it necessary to discuss this in this particular section of the manuscript any further.

102: Why was a reduced set of populations used?

Because both genotype and phenotype information was available only for the four populations used in this study. That is, we had more genotype than phenotype data.

185: Specify here what averaging method was used to combine information about F_{ST} across loci. Later (line 411) it becomes clear that a biased form of averaging was used. It is well known that averaging the point estimates of F_{ST} of multiple loci gives a biased estimate of the F_{ST} produced by the underlying biological process. Weir and Cockerham showed that the method of taking the ratio of the average numerator and average denominator for F_{ST} across loci is much less biased. Results based on an estimation method already known to be flawed won't be helpful.

We agree with the reviewer that our original choice, average of ratios (AOR), was not an ideal method for combining the information across loci although it appears as the most natural approach to describe the changes in sitewise distributions shown in Fig 1. We have now expanded the analyses, incorporating the two averaging approaches (AOR and Ratio Of Averages [ROA]) as well as Reviewer 2's comments about marker ascertainment.

Figure 2 b+c are not necessary.

Given the new expanded analyses contrasting full data with population-based ascertainment of loci, we believe that the two figures are very useful.

284: What does it mean that the "increase was consistent with the proportion of sites discarded"? As far as I know there is no theoretical prediction on this to say that the effect was consistent.

We have modified that part of the manuscript significantly and the statement has been removed.

298: "neither" -> "either"

Fixed

375: I appreciate that the authors are saying that their results for sticklebacks should not be considered a general rule. But this is a bit of a strawman—the results from this one case really don't inform us about the general pattern, and that is the whole context of the paper.

The reviewer is in the mindset that we use a "strawman" argument here. As pointed out above, we set out to make an empirical test of the prediction (cf. Edelaar & Björklund 2011, Edelaar et al. 2011) that the use of microsatellites makes Q_{ST} - F_{ST} comparisons overly liberal. As the results show, in this particular case the outcome in terms of Q_{ST} - F_{ST} tests was that the choice of marker type did not make a difference to the outcome. However, detailed analyses of the marker variability show that the predicted bias towards deflation of F_{ST} is present when using microsatellites. Likewise, the way the microsatellite and SNP data are pruned and ascertained influence neutral baseline levels of differentiation. To us, these are all valid conclusions based on the data.

404: I believe the authors mis-cite reference 72. This paper is about ascertainment bias, not about

minor allele frequency filtering. There is truly no real controversy about whether lower MAF loci give biased estimates of F_{ST} .

We have removed this mis-citation.

464: It is not surprising that increasing the number of loci for F_{ST} -estimation did not help, because the main source of sampling error in most Q_{ST} F_{ST} comparisons is uncertainty in Q_{ST} , not F_{ST} .

We agree with the reviewer that Q_{ST} is the main source of uncertainty in Q_{ST} - F_{ST} comparisons. However, the discussion over how many markers are needed to obtain confident F_{ST} estimates is relevant – at least when considering microsatellites. In this perspective, our results provide potentially valuable information.

A paper that perhaps ought to be cited is Wang's paper on correcting bias for F_{ST} with microsatellites <https://www.ncbi.nlm.nih.gov/pubmed/25891752>.

Thank you – cited now.

Reviewer 2:

General Comments

This manuscript takes up an interesting question about Q_{ST} - F_{ST} comparisons used to test for selection on quantitative traits. In particular, the F_{ST} estimate often depends on aspects of genetic marker type and marker selection. (Method of estimation matters too but is not considered in the manuscript; all analyses focus on Weir-Cockerham estimation, which is the consensus choice.) The effects of marker type and marker selection on F_{ST} have been considered in many contexts, but the authors note that it has received less attention in the Q_{ST} - F_{ST} literature. Overall, I felt that the claims made in the manuscript were justified by the data presented, and the scripts used for data analysis and generation of simulation data have been made available. Most of the below is presented in the spirit of showing the authors my constructive reactions to the work, and I do not think that they all need responses in order to meet the objective peer review criteria of Royal Society Open as I understand them.

We thank the reviewer for these positive comments and particularly appreciate the effort made in his review which was, indeed, very constructive.

The authors pursue questions about F_{ST} estimation primarily in a dataset of several stickleback populations and also in a limited simulation study inspired by stickleback demography. (I felt that the simulation arm of the study was not used as fully as it might have been and also that it could have been larger to give clearer resolution on the results.)

We agree with the reviewer about this comment, which is also in line with Reviewer 1's opinion on the matter. Due to the computational constraints of the Q_{ST} - F_{ST} methods used in the study – specifically the MCMC-based driftsel – simulations became rapidly prohibitive in terms of computation time. Please see our response to Reviewer 1 as well as our detailed response below.

In general the authors report changes to estimated F_{ST} values that are sensible and expected given prior literature, but these variations do not cause much difference in the reported QST- F_{ST} comparisons. Presumably, the simulation framework could be used to find parameter regimes where the choices matter more.

One issue that puzzled me while reading the manuscript was not the fault of the authors but arises from the long history of F_{ST} in population genetics. As the authors know, F_{ST} has been variously interpreted as (i) a parameter of a data-generating process (e.g. of an evolutionary model of two or more subdivided populations) or as a function of those parameters and (ii) as a summary statistic describing genetic differentiation (e.g. as a proportion of variance).

Thank you for this insightful comment. We now discuss this point in the main text (see L135 onwards in the main text)

If one views F_{ST} as a parameter, then standard F_{ST} estimators for microsatellites (e.g. GST-style estimators) are often viewed as biased downward (e.g. Whitlock, 2011). If, on the other hand, one views F_{ST} as a descriptive statistic, then it is generally acknowledged to be smaller for more variable markers (e.g. Holsinger & Weir, 2009). QST- F_{ST} comparisons can be viewed from either point of view. The $QST \sim F_{ST}$ result has been derived many times independently, but in the derivations with which I am most familiar, F_{ST} is generally viewed either as (i) a function of parameters that is estimated accurately by low-mutation-rate markers (e.g. Whitlock 1999; Koch 2019 Genetics) or (ii) a statistic describing neutral differentiation specifically at biallelic markers (usually because the loci affecting the trait are taken to be biallelic; e.g. Berg & Coop 2014; Edge & Rosenberg 2015 Human Biology). I don't think that using a Weir-Cockerham F_{ST} computed from microsatellites makes sense under either view---under view (i) it is biased downward as an estimate of the parameter of genuine interest, and under view (ii) it is a description of differentiation at the wrong class of markers, which is unlikely to match differentiation at the right class of markers. It is possible that an alternative estimator developed for microsatellites, such as Slatkin's (1995) RST, could be a good estimator of the parameter of interest under view (i), as long as its assumptions accurately reflect the markers' history (e.g. for RST, the stepwise mutation model would have to be a reasonable fit). In some ways, these issues are orthogonal to the manuscript, which asks whether using standard F_{ST} estimators affects the QST- F_{ST} comparison empirically. The answer seems to be that it easily could do so in some cases but does not for the specific empirical data in hand. That said, I think that some discussion of these points would clarify the statistical framework that is implicit in QST- F_{ST} comparisons, and I do think that comparison with other microsatellite-based estimators such as RST would enrich the story here. Another discussion point I would have liked to see was more talk about marker ascertainment. In systems with less fully developed genomic resources than sticklebacks, marker ascertainment may lead to an overrepresentation of highly heterozygous markers. There are already data on this in the manuscript, but I did not recall comments on this specific view of the issue.

We appreciate these interesting reflections. Microsatellite markers have been widely used--and debated--in \$F_{ST}\$ literature. We agree that a highly mutable microsatellite locus may not correctly represent frequency changes of ancestral alleles and two alleles can be identical by state due to independent mutations. Moreover, it is difficult to see how any statistics could correctly estimate processes that are not significantly faster than the marker mutation rate. However, if the mutation rate is very slow, we do not see why \$F_{ST}\$ estimates based on microsatellites would be much different from those based on SNP data. We use microsatellite allelic richness as a proxy for

mutation rate and show that F_{ST} estimates for the lowest category are indeed most similar to those of SNP data. We have now clarified this argument.

We did not fully understand the reviewer's suggestion about marker ascertainment. If the comment referred to ascertainment of microsatellite markers, we have now tried to clarify the conclusions from this part of the analysis and the corresponding discussion. If it referred to population-based ascertainment, we have now followed the recommendations of Bhatia et al. 2013 and studied the impact of ascertainment by variability in the two most diverged lineages in our data (see L133 onwards). Combined with Reviewer 1's comments about averaging sitewise estimates, the results and the corresponding conclusions can change quite significantly depending on whether ascertainment is done or not. If the comment referred to other marker types, we note that our study is based on full genome data only, and we do not see it relevant to discuss other genotyping approaches in this context.

Specific Comments

"However, since per nucleotide mutation rates are highly variable, and may vary depending on their genomic location [20, 21], SNP loci may not automatically yield less biased neutral baseline estimates of divergence than microsatellites"

-Biased as an estimator of what in this case?

We meant biased as an estimator of F_{ST} , representative of the neutral baseline in the case of QST- F_{ST} comparisons. We clarified the sentence, which now reads: *"SNP loci may not automatically yield less biased neutral baseline estimates of divergence (as measured by F_{ST}) than microsatellites"*

"this would be indicative of uniform stabilizing selection that has prevented populations from diverging less than would be expected under random genetic drift alone"

The phrasing is confusing. Perhaps changing "less than" to "as much as" would help.

We rephrased the sentence, which now reads *"this would be indicative of uniform stabilizing selection, with populations being less divergent than expected by random genetic drift alone"*

"Using the inferred rate, the SNP dataset NONCOD and the microsatellite data were split into subsets of loci with low (1-7 cM/Mb) and high (7-15 cM/Mb) local recombination rates, discarding the loci with either no rate estimate or an estimate from the extreme ends of the rate distribution (Fig. S3a). The microsatellite loci were further divided to those with few (1-4), intermediate (5-8) and many (9- 21) alleles (Fig. S3b), and F_{ST} was separately estimated for the different subsets"

-Is there a justification for these specific bins (e.g. are these median or tertile splits)? Any justification for these bins?

We have now clarified the procedure in the text. With microsatellites, our main aim was to compare the two extreme categories and they approximately correspond to the first and fourth quartile (given the discrete nature of the data). With recombination rates, the two extremes are potentially erroneous and may reflect errors in the underlying data; we chose to include two roughly equally large subsets from the middle of the distribution. We admit that the choice of these bins is indeed slightly arbitrary but we do not expect this choice to influence the inference.

-I would be more curious to see results for binning msats by expected heterozygosity rather than by numbers alleles. As shown in the supplement, these are correlated, but heterozygosity has a direct link to F_{ST}.

With highly mutable markers, the link between heterozygosity and F_{ST} is slightly unclear. Moreover, we believe that, for a small number of samples, the allelic richness is likely to capture better the differences amongst the loci. We decided to skip this suggestion, as it is unlikely to lead any fundamental changes in the main inference of the manuscript.

-Are these the total numbers of alleles or the number of non-reference alleles? (And if total, why are there some msats with 1 allele?)

That is the total number of alleles. Initially, all loci showing perfect repeat of required length in the reference genome were included and a small number of loci may then have not shown variation in any of the samples. Naturally, only variable loci are used in later analyses.

It would help the reader to give a few sentences more about the differences between the driftsel model and standard QST-FST comparisons. In particular, the sentence "it can detect selection even when QST = FST" is confusing without further details given the rationale for QST-FST comparisons explained in the introduction.

We agree with the reviewer. However, because the statistical framework underlying the Driftsel procedure is fairly advanced, we choose to direct the readers to the relevant publication for a detailed explanation on the approach, and removed the sentence pointed out by the reviewer.

"Simulations were based on the most likely demographic history of real nine-spined stickleback populations"

-In what sense is this demographic history "most likely"?

The demographic history of the nine-spined sticklebacks in this region has been inferred in the past using mitochondrial and nuclear genetic markers (see references 59-61 in the manuscript). As such it represents the most likely scenario of colonization of the species in northern Europe (as opposed to the real history which is unknown). Thus, we modeled our simulation scheme following these results.

"Each chromosome contained four LD blocks: one comprising of 250 SNPs, and the other three of 25 microsatellites each"

-Why are all SNPs on a chromosome in LD but not in LD w/ the msats?

This is mainly due to the limitation of the software "fastsimcoal2", which could not simulate LD between different types of markers. However, this should not affect the results, since we conducted the FST calculation on different LD blocks separately.

"In scenario (ii) to (iv), populations were subjected to increasing levels of directional selection (QST > FST) by multiplying the neutral pattern by a factor of 1.5, 2 or 4, respectively, thus generating weak (ii), moderate (iii) and strong selection (iv)."

-What does "multiplying the neutral pattern" mean?

This was initially introduced in the article Karhunen et al. (2013) as a way to simulate a scenario where Q_{ST} is larger than F_{ST} . It means to multiply the population mean of phenotypes (expectation under neutrality) to make the mean larger than expectation. For more details, please refer to the supporting appendix 2 in Karhunen et al. (2013) driftsel: an R package for detecting signals of natural selection in quantitative traits. *Molecular Ecology Resources*. 13, 746-754.

“The whole simulation procedure was replicated 10 times, and the averaged performances of both Driftsel and QstFstComp on the replicated data sets were recorded.”

-Would it be possible to increase the number of simulations? 1000 would be best but might be inaccessible—if so, I would suggest giving the reader a sense of how long these simulations take. At least 100 would give a lot more precision for these results.

Unfortunately, the simulation procedure took substantial amount of time to be completed (3 months). For example, a single MCMC run of Driftsel will take about one day for 80 individuals and 2000 SNPs (we now state this in the main text as suggested by the reviewer). Therefore running 1000 replicates would take months. However, we are confident that the results from 1000 simulations replicates would not be different from those based on 10 replicates. This because the results we obtained agree very well what the theory predicts. We now give the reader a sense of computation time constraints L310.

“Our results indicate that microsatellite markers are indeed likely to yield an overly liberal neutral baseline for Q_{ST} - F_{ST} tests,”

I think this sentence is stated a little too strongly given that the authors have not identified cases where it does matter.

We have now modified this sentence to better reflect what can be said on the basis of our results: *“.. our results indicate that microsatellite markers can indeed – unless carefully filtered - yield an overly liberal neutral baseline for Q_{ST} - F_{ST} tests”.*

I found myself somewhat confused by the explanation of the pattern of F_{ST} results produced by the LD thinning. The explanation seems to turn mostly on both (a) the very different within-population LD in different populations and (b) the pruning of sites with fixed or near-fixed differences in the pooled data. I am missing what (a) has to do with (b): the fixed or nearly-fixed sites presumably should have little LD w/ neighboring markers within populations, so I don't see how the different levels of within-population LD matter. It seems to me that if you have fixed differences, then those will tend to show up as being in high LD (across populations) with other fixed differences, so if there are many fixed differences in a region, then many will be pruned on the basis of pooled data. If pruning affects these markers more strongly than it does markers that are less differentiated, then F_{ST} might be biased downward.

You are right. We now better discuss the effect of LD-pruning on F_{ST} estimates in the main text (L489-513).

This review was written by Michael D. Edge.

Thank you for being up front about your identity.